# Drought-Induced Civil Conflict Among the Ancient Maya

Douglas J. Kennett [1✉], Marilyn Masson[2], Carlos Peraza Lope[3], Stanley Serafin[4], Richard J. George[1], Tom C. Spencer[5], Julie A. Hoggarth[6], Brendan J. Culleton[7], Thomas K. Harper [8], Keith M. Prufer [9,10], Susan Milbrath[11], Bradley W. Russell[2], Eunice Uc González[3], Weston C. McCool[12], Valorie V. Aquino[9], Elizabeth H. Paris [13], Jason H. Curtis [14], Norbert Marwan [15], Mingua Zhang [16], Yemane Asmerom [17], Victor J. Polyak [17], Stacy A. Carolin[5], Daniel H. James[5], Andrew J. Mason[18], Gideon M. Henderson[18], Mark Brenner[14], James U. L. Baldini[19], Sebastian F. M. Breitenbach [20] & David A. Hodell [5✉]

The influence of climate change on civil conflict and societal instability in the premodern world is a subject of much debate, in part because of the limited temporal or disciplinary scope of case studies. We present a transdisciplinary case study that combines archeological, historical, and paleoclimate datasets to explore the dynamic, shifting relationships among climate change, civil conflict, and political collapse at Mayapan, the largest Postclassic Maya capital of the Yucatán Peninsula in the thirteenth and fourteenth centuries CE. Multiple data sources indicate that civil conflict increased significantly and generalized linear modeling correlates strife in the city with drought conditions between 1400 and 1450 cal. CE. We argue that prolonged drought escalated rival factional tensions, but subsequent adaptations reveal regional-scale resiliency, ensuring that Maya political and economic structures endured until European contact in the early sixteenth century CE.

[1] Department of Anthropology, University of California, Santa Barbara, CA 93106, USA. [2] Department of Anthropology, The University of Albany-SUNY, Albany, NY 12222, USA. [3] Instituto Nacional de Antropología e Historia – Centro INAH Yucatán, Mérida, Yucatán 97310, Mexico. [4] School of Medical Sciences and Earth and Sustainability Science Research Centre, University of New South Wales, Sydney, NSW 2052, Australia. [5] Godwin Laboratory for Palaeoclimate Research, Department of Earth Sciences, University of Cambridge, Cambridge CB2 4EQ, UK. [6] Department of Anthropology & Institute of Archaeology, Baylor University, Waco, TX 76798, USA. [7] Institutes of Energy and the Environment, The Pennsylvania State University, University Park, PA 16802, USA. [8] Department of Anthropology, The Pennsylvania State University, University Park, PA 16802, USA. [9] Department of Anthropology, University of New Mexico, Albuquerque, NM 87106, USA. [10] Center for Stable Isotopes, University of New Mexico, Albuquerque, NM 87106, USA. [11] Florida Museum of Natural History, University of Florida, Gainesville, FL 32611, USA. [12] Department of Anthropology, University of Utah, Salt Lake City, UT 84112, USA. [13] Department of Anthropology and Archaeology, University of Calgary, 2500 University Dr. NW, Calgary, Alberta T2N 1N4, Canada. [14] Department of Geological Sciences, Land Use and Environmental Change Institute, University of Florida, Gainesville, FL 32611, USA. [15] Potsdam Institute for Climate Impact Research, Member of the Leibniz Association, Potsdam 14473, Germany. [16] Stony Brook University, Stony Brook, NY 11794, USA. [17] Department of Earth and Planetary Sciences, University of New Mexico, Albuquerque, NM 87106, USA. [18] Department of Earth Sciences, Oxford University, Oxford OX1 3AN, UK. [19] Department of Earth Sciences, University of Durham, Durham DH1 3LE, UK. [20] Department of Geography and Environmental Sciences, Northumbria University, Newcastle upon Tyne NE1 8ST, UK. ✉email: kennett@anth.ucsb.edu; dah73@cam.ac.uk

Archeological and historical studies have proposed linkages among global climate change, societal instability, violent conflict, and sociopolitical collapse[1–7], but also instances of resilience, transformation and sustainability in the face of climate pressures[8–10]. The influence of climate change on civil conflict in the last century has also been the focus of compelling statistical studies[11,12], and an important nexus for debate, revealing the importance of human agency and unexpected, non-linear relationships between climate and human behavior[13–19]. Longer-term climatic, archeological, and historical records can contribute to these contemporary debates, but demand a rigorous transdisciplinary framework that bridges natural and social systems[8,18]. We report a singular case study of the complexities of the natural and social systems at the Postclassic Maya capital of Mayapan (1200–1450 calendar year [cal.] CE) throughout its history and its ultimate demise. This occurred in the context of drought, civil conflict, and the collapse of the regional state. A period of political balkanization after 1450 cal. CE preserved complex organizational institutions and fostered a resilient, peninsular-wide market economy observed at European contact in the early 16th century CE. We examine climate stresses to Mayapan's local and regional subsistence and economic systems, and the behaviors of human actors that included political violence within these dynamic social and political transformations.

Climate assessment reports of the intergovernmental panel on climate change (IPCC) evaluate risk and model vulnerability during the last century at regional and global scales[20]. Long-term climate reconstructions reveal that some climate variations in the past were of significantly greater magnitude than those experienced in the last 100 years. Furthermore, current and future anthropogenic influences are projected to amplify the severity of extreme events in the water cycle and lead to more intense and prolonged droughts than those that impacted agricultural productivity in the recent past[21,22]. Archeological and historical records are well-suited for examining past societal effects of climate crises over long-term cycles. Such effects in Pre-Columbian Mesoamerica may have included migration, demographic decline, geographic shifts in centers of political power, and warfare. We focus on the latter, specifically the initiation and consequences of societal conflicts that coincided with climate impacts on the agrarian food base and regional political economy. These subsistence and related economic foundations are well-known from the archeological record and from memories of descendant peoples in early Colonial period European accounts[12]. The magnitude and duration of the climate impacts in the northern Yucatán Peninsula (Mayapan) may have transcended established mechanisms to overcome modest inter-annual rainfall fluctuations[23], given the limitations of cumbersome transport and the long-term storage of maize, the primary staple grain[2,24].

The Maya region offers the breadth and depth of archeological, historical, and climate records essential for studying correlations between social change and fluctuating climate conditions. Here, we compare the archeological and ethnohistorical evidence for civil conflict with local and regional records of climate change at the last large Pre-Columbian Maya capital city of Mayapan (1200–1450 cal. CE, Fig. 1). Multiple studies have shown that patterns of political formation, consolidation, and fragmentation of Maya states were cyclical well before Mayapan became a Postclassic Period political center. From the middle of the first millennium BCE until Mayapan's emergence, kingdoms formed, consolidated, or expanded their domains of subjects, then ultimately fragmented at the same time that new centers emerged to dominate the landscape[25–28]. Large cities first appeared along the western edges of the Maya region by 1000–800 cal. BCE[29] and numerous monumental centers and networks of villages developed in the heart of the Maya Lowlands by 500 cal. BCE[30]. One of the largest state capitals in all of Maya prehistory (in terms of monumentality), El Mirador, was established between 100 cal. BCE and 250 cal. CE[27]. The Lowlands region was dotted with kingdoms large and small, amidst a densely populated countryside of towns and villages throughout the Classic period, with the Late Classic Period (600–800 cal. CE) witnessing the emergence of the largest number of competing states in the southern lowlands. Shifting centers of influence and the most dramatic sequential disintegration of these cities occurred during the Terminal Classic period, between 800 and 1000 cal. CE, with climate change argued to be a contributing factor to increasing conflict[2,31–34]. During this interval, the towns and cities of the northern and coastal peninsula (e.g., Uxmal, Chichen Itza, Aventura) were magnets for Maya settlement, in turn experiencing political and demographic decline that coincide with multiple lines of evidence for drought during the eleventh century cal. CE[10,35,36]. But northern and coastal Maya populations subsequently recovered quickly and the political capital of Mayapan arose and governed from at least 1100 to 1450 cal. CE[37]. Mayapan provides an appropriate test case for exploring the causes and impacts of civil conflict because of its temporal proximity to European Contact, and retrospective, Colonial-Period documentary accounts provide additional evidence that can be evaluated with chronological (radiocarbon [14C] dates) and human osteological data.

Mayapan emerged as a regional capital on the Yucatán Peninsula, following the demise of Chichen Itza between 1000 and 1100 cal. CE[38,39]. Numerous political families, smaller polities, and substantial populations endured through the fall of the Chichen Itza polity. Many of these entities were reintegrated into the Mayapan confederacy, especially those concentrated in the northwest region of the peninsula[35]. Historical sources indicate that the most influential nobility came from the houses of the Cocom, the Xiu, and the Chel (among others) who governed the polity as members of Mayapan's ruling council[40,41]. These lords established the city's monumental center, replete with its principal pyramid, the Temple of K'uk'ulkan, and a nucleated set of other temples, colonnaded halls, and shrines covered in murals and sculptures that reflect the city's mythical foundations[39,42]. Densely settled residential zones extend from the center in all directions within the city's 9.1 km-long circumferential wall, which encloses an area of 4.2 km², and housing sprawls at least a half kilometer beyond this boundary (Fig. 1)[43]. Twelve formal gates in the wall directed pedestrian traffic into and out of the city; the wall was clearly a defensive feature[43]. The monumental and settlement zones were founded with the intent of establishing a new political capital. Population aggregation and recruitment across the Yucatán peninsula, and occasionally beyond, persisted throughout Mayapan's history as indicated by strontium isotopes in human teeth[44]. Subject peoples were summoned to move to the city to provide all manner of services[41]. Human remains are ubiquitous at the archeological site, which was once home to 15–20,000 inhabitants who were sustained by household gardens and orchards, hunting, and rain-fed maize agriculture, supplemented by trade[37]. Dietary stable isotope studies indicate heavy reliance on maize, a crop that was highly sensitive to periodic droughts, given the limitations for long-term grain storage[44,45].

In this work, we use a transdisciplinary approach that combines archeological, historical, and paleoclimate datasets to examine the dynamic interactions of climate change, civil conflict, and political collapse at Mayapan during the 14th and 15th centuries CE. We use generalized linear modeling to demonstrate that increases in traumatic injuries evident in the bioarchaeological record strongly correlate with drought conditions between 1400 and 1450 cal. CE. We argue that prolonged drought escalated rival factional tensions that ultimately resulted in the

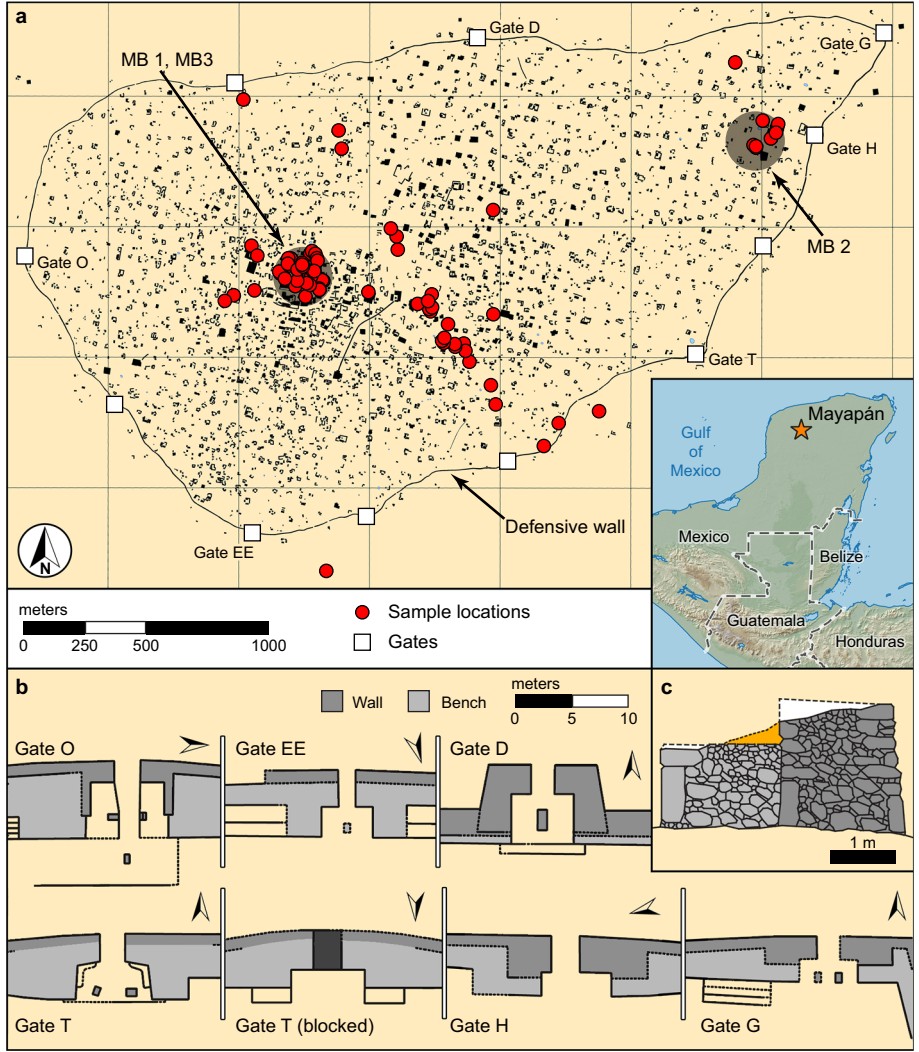

**Fig. 1 Map of Mayapan. a** Layout of the city showing housing complexes, the defensive wall, formal gates, and the locations of skeletal samples (red dots). **b** Gate and wall configurations adapted from[43]. **c** Cross-section of the tiered double-wall construction surrounding Mayapan. MB, Mass Burial.

abandonment of the city, but that Maya political and economic structures endured regionally until European contact in the early 16th century CE.

## Results

**Integration of archeological and historical observations**. We used accelerator mass spectrometry (AMS) to $^{14}$C-date purified bone collagen from 205 human skeletons to develop a high-temporal-resolution history of population fluctuations at Mayapan (Fig. 2c; also see Methods and Supplementary Note 4). We then integrated these data with archeological, historical, osteological, and paleoclimatological data to explore relations between periods of societal stability or fragility, and drought. These data indicate a rural population in the vicinity of this location from the Middle Preclassic (~1000 cal. BCE) through the Colonial Period. After the establishment of the Postclassic capital by 1150 cal. CE, there was rapid nucleation of populations starting ~1200 cal. CE that continued through 1320 cal. CE (Fig. 2c).

According to retrospective historical documents, Mayapan was a militaristic state that raided its peninsular enemies and took war captives destined for slavery or sacrifice. Historical records can be biased, so we used archeological, osteological, and radiocarbon data to establish the timing of a series of events recorded historically using the Maya K'atun calendar[4,40]. These intervals

include: 1) a period of "terror and war" [1302–1323 cal. CE], 2) an interval of "inquisition against members of the Xiu nobility" [1361–1381 cal. CE]; 3) decentralization and population decline [1382–1401 cal. CE], 4) massacre of the Cocom noble family [1440–1461 cal. CE] and 5) political demise of the regional state and abandonment of the city [post–1450 cal. CE] (Fig. 2a–d).

**Incorporating regional climate & paleoclimate records**. To investigate linkages between these events and climate, we compared the records of cultural change to local and regional paleoclimate records. Modern rainfall on the northwestern Yucatán Peninsula is marked by a strong north-south gradient, from a minimum of 450 mm yr$^{-1}$ at Progreso on the northwest coast, to 1000 mm yr$^{-1}$ at Mérida, and 1150 mm yr$^{-1}$ at Abala, near Mayapan (Supplementary Fig. 1). A pronounced west-to-east increase in precipitation across the northern peninsula, a consequence of the prevailing easterly winds heading inland (westward) from the Caribbean water source, also exists. Rainfall is highly seasonal, with the wet season lasting from May to October. Northern Yucatán lies within the main North Atlantic hurricane belt and a single storm event can contribute a substantial proportion of annual precipitation for that year. Farming in the region depends heavily on the timing of the onset,

duration, and total effective rainfall of the wet season[46]. Although short-term droughts tend to be spatially and temporally patchy in northern Yucatán[47], more protracted droughts during the historic period are evident in documentary and paleoclimate archives[48,49].

Those extended dry intervals of the Colonial Period caused well-documented crop failures, severe famines, and high mortality that destabilized the economy and led to periodic dispersal of populations from towns in the northern Yucatán[48].

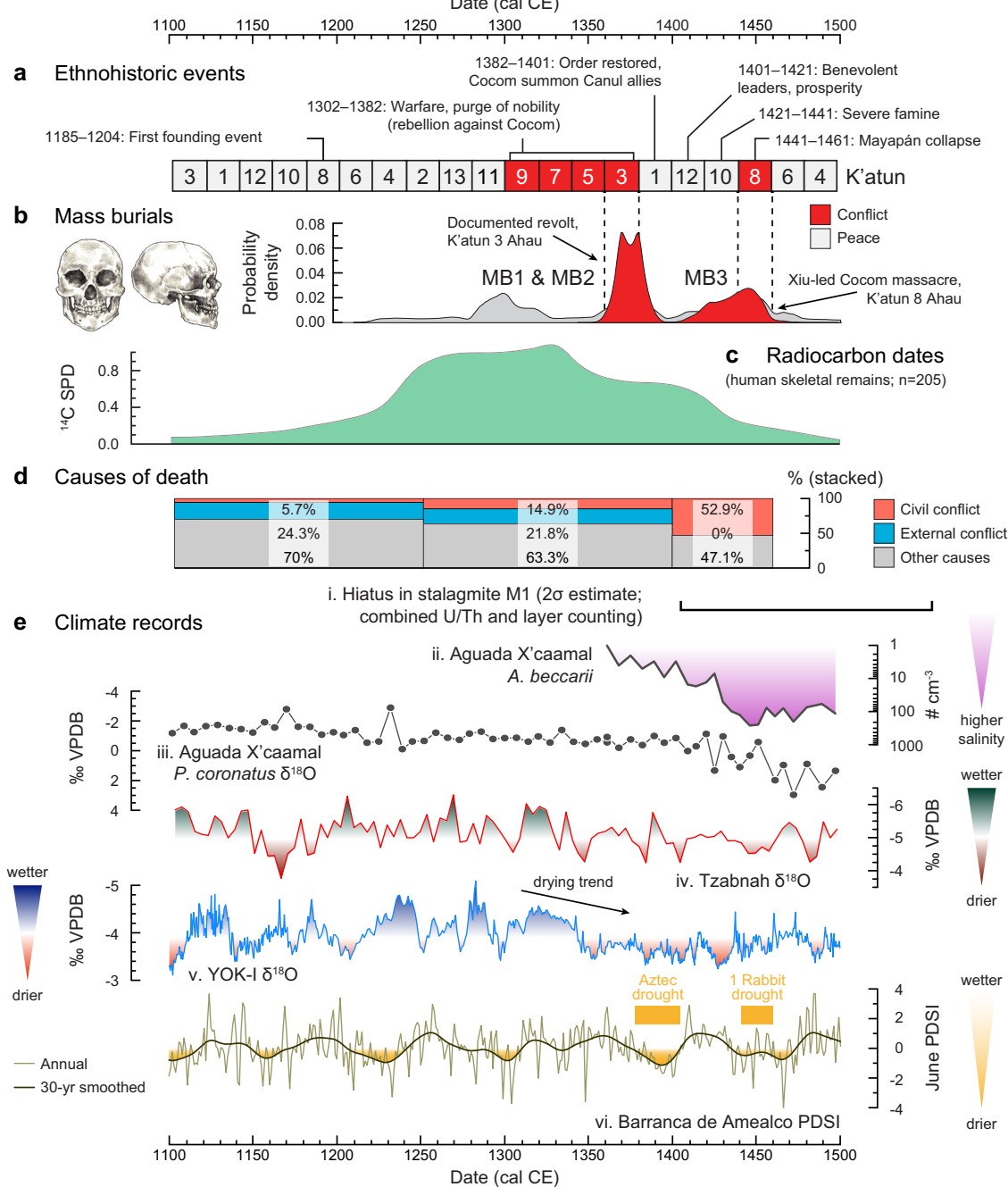

**Fig. 2 Culture and climate records. a** Retrospective events recorded historically using the Maya K'atun calendar, highlighting intervals of conflict and peace within Mayapan[40,41]; **b** Summed probability distributions of AMS-[14]C-dated human skeletons from mass burials MB1, MB2, and MB3 as an independent measure of internal conflict (OxCal Code and model provided in Supplementary Code 2); **c** Summed probability distribution of directly AMS-[14]C-dated human skeletal remains from Mayapan as a population size estimate[88] (see methods and Supplementary Discussion 4; Code available in Supplementary Code 1); **d** Osteological data indicative of internal or external conflict and other causes. The sample included 35 individuals <18 years of age, 142 individuals ≥18 years of age (of which 48 are males, 49 are females and 45 are adults of indeterminate sex), and 28 individuals of unknown age (see Supplementary Discussion 3 and Supplementary Data 1). **e** Local and regional climate records. (i) Estimated age (± 2σ) of the hiatus for Mayapan stalagmite M1 (this study). (ii) Abundance of the benthic foraminifer *Ammonia beccarii* in the sediment profile from Aguada X'caamal[90] with an updated time scale (this study). (iii) Oxygen isotope record of the snail *Pyrgophorus coronatus* from the same core[90]. (iv) Oxygen isotope record from the Chaac speleothem from Tzabnah Cave, Tecoh, Yucatán[53]; (v) Oxygen isotope record from stalagmite Yok-I from Yok Balum Cave, Belize[2]; (vi). Palmer Drought Severity Index (PDSI) inferred from tree ring record from Barranca de Amealco, Mexico[61]. All unpublished datasets plotted are provided in Supplementary Data 2 and the Source Data File. U/Th, Uranium/Thorium, MB, Mass Burial.

Rainfall over the Yucatán Peninsula is derived mainly from the Caribbean, Atlantic, and Gulf of Mexico sources, and only occasionally from the Pacific[2]. Summer wet season rainfall dynamics on the peninsula are influenced by the Caribbean Low-Level Jet (CLLJ), an easterly trade wind system. Variability of the CLLJ depends on the position and strength of the Bermuda High[50], meridional sea level pressure (SLP), and sea surface temperature (SST). Stronger trade winds, caused by a high meridional SLP gradient, result in lower SSTs and drier conditions on the Yucatán Peninsula. The sea-level pressure gradient between the Caribbean and the eastern Pacific links the region in a complex fashion with the inter-tropical convergence zone (ITCZ) and the El Niño-Southern Oscillation (ENSO) system in the Pacific[51,52], influencing Yucatán moisture dynamics indirectly. Such linkages tend to occur in the wet season, prior to El Niño events[51], which reduce tropical depressions and hurricanes in the study region. Outside the rainy season, winter *nortes* occasionally bring rainfall and cold, windy weather from the Gulf of Mexico[46].

The YOK-I speleothem $\delta^{18}O$ record[2] shows a series of wetter intervals between 1100 and 1340 CE, an interval when populations at Mayapan were growing (Fig. 2e). A drying trend that began after 1340 CE is inferred from YOK-I $\delta^{18}O$ data and is consistent with the more local, but lower-resolution Chaac speleothem $\delta^{18}O$ record from Tzabnah Cave in Tecoh, located ~12 km north of Mayapan (Supplementary Note 1)[53,54]. The mid-14th-century drying trends are consistent with a growth hiatus between 1397 and 1427 CE ($2\sigma$ confidence interval) identified here in a stalagmite (M1) collected from a cave located directly below the central Plaza at Mayapan, and analyzed for the present study (Supplementary Notes 1 and 2; Supplementary Data 1). One explanation for cessation of growth is that the climate became so dry at Mayapan that drip water stopped reaching the stalagmite and calcite deposition ceased. When deposition resumed about 1600 CE, the stalagmite mineralogy changed from calcite below to aragonite above the hiatus. At Aguada X'caamal, a small sinkhole ~27 km west of Mayapan, the $\delta^{18}O$ and salinity of the lake water increased abruptly at 88 cm in the core, corresponding to a date of 1345–1458 CE ($2\sigma$ confidence interval; Supplementary Note 2). Stable isotope and faunal changes in Aguada X'caamal coincide approximately with the detected hiatus and mineralogical transition from calcite to aragonite in stalagmite M1 from beneath Mayapan.

**Population Dynamics and Civil Conflict**. Summed probability distributions (SPD) of directly radiocarbon-dated human skeletal material (Fig. 2c, Supplementary Fig. 14, Supplementary Note 4) from Mayapan show steady population growth and nucleation starting ~1100 cal. CE and a peak between 1200 and 1350. Populations started to decline after ~1350 cal. CE and were very low by 1450 cal. CE. This is consistent with both the K'atun 1 Ahau (1382–1401 cal. CE) depopulation and the K'atun 8 Ahau (1441–1461 cal. CE) depopulation and abandonment of Mayapan, mentioned in multiple historical sources (e.g., Fray Diego de Landa and the Books of Chilam Balam of Chumayel, Tizimin, and Maní)[40]. Public artistic programs and historical records also suggest greater interaction with Aztec populations from Central Mexico during that time, initially through Mexican mercenaries and traders[55]. At the same time, the Cocom lineage aligned itself with Canul allies at the end of the 14th century, associated with internal disputes with their Xiu rivals (Fig. 2a)[40].

Mayapan's cityscape contains varied and numerous funerary deposits, as well as extra-funerary deposits, interpreted as mass graves (Supplementary Note 3; Supplementary Data 2). Multiple individuals were often interred together in household graves, and ossuaries containing the secondary deposits of skeletal material exist within the monumental center, as well as in high-status houses in the residential zone[56–58]. In contrast, other large deposits of human remains in the monumental center (e.g., Round Temple Complex) likely represent sacrificed (or otherwise killed) war captives from polities targeted by the Mayapan state elsewhere on the peninsula (beyond the northwest stronghold of the confederacy). One such deposit was found in an alleyway next to the site's iconic Round Temple, which contained the crania and long bones of 25 individuals who were directly AMS-[14]C-dated to 1302–1362 cal. CE (Supplementary Note 3). Three of the individuals suffered perimortem cranial trauma and cut marks on some of the bones, indicating postmortem dismemberment and defleshing, likely caused by intentional desecration[56,57,59]. These mortuary deposits may correspond to historical references of conflict and strife waged on the Yucatán Peninsula between 1302 and 1362 cal. CE[40], during a period when Mayapan's monumental landscape was fully built and when its population and regional influence were at their peak.

Another type of mass grave in the city is unlike either funerary features or deposits of the remains of war captives. Rather, it is indicative of violence experienced by individuals who were likely high-status residents of Mayapan. These two shallow mass graves also exhibit desecration of human and ritual pottery remains (deity effigies); both are adjacent to ceremonial edifices. Both features date to the late 14th century (1350–1400 cal. CE; Supplementary Note 3). One is in the center of the city by Temple Q-80 (MB1; Q.79; 10 individuals; Supplementary Fig. 9) and the other is next to the Itzmal Ch'en temple and colonnaded hall, part of an outlying ceremonial group near the far eastern gates of the city wall (MB2; 20 individuals; see Fig. 1)[37,57]. Both MB1 and MB2 are shallow graves that contain multiple, mostly disarticulated individuals who were buried along with smashed pottery (Chen Mul Modeled censers). Four of the individuals in MB1 suffered violent deaths, as indicated by stone knives embedded in their scapula, rib cage, or pelvis[60]. The bones of ~20 individuals in MB2 were chopped into pieces and burned. Both observations are consistent with historical records for the late 14th century CE, which indicate that one faction of the government persecuted, tortured, and executed rival factions. Following this turmoil, sources suggest a restoration of order and several decades of relative stability just before and after 1400 CE[40]. Direct AMS [14]C dates on 10 individuals from MB1-MB2 are statistically indistinguishable from one another, with a modeled age of 1360 to 1400 cal. CE. This coincides with a major central Mexican drought in the Aztec-controlled highlands (1378–1404 cal. CE)[61], which was the most extreme dry episode since 771 cal. CE. The historical records of massacres at Mayapan also correspond with the population decline that is evident in the decreasing number of directly AMS [14]C-dated individuals (Fig. 2c), as well as evidence for the destruction and abandonment of some buildings around the same time as MB1 and MB2[37]. Remnant populations persisted in the city, but new architectural construction was much reduced as violence increased.

The last recorded mass burial at Mayapan (MB3) flanks the northern staircase of the Temple of K'uk'ulkan, and is the best candidate for the Xiu-led massacre of members of the Cocom family that resulted in the city's collapse and ultimate abandonment. The skulls and isolated limbs of nine individuals, many of them children ($n = 7$), were found mixed with debris from building collapse (Supplementary Note 3). Two of these individuals show evidence for violent death in the form of

perimortem rib puncture wounds, consistent with a massacre. Direct AMS-[14]C measurements indicate that they were killed after 1400 cal. CE. Radiocarbon dates for six of these individuals are statistically indistinguishable and suggest a single massacre between 1440 and 1460 cal. CE. Mitochondrial DNA indicates that these individuals shared strong maternal genetic affinities and were potentially related on the maternal line[44]. Remarkable agreement between the historical accounts of this politically charged, divisive event and the calibrated radiocarbon dates on physical remains suggests that MB3 pertains to the violent battle and destruction of Mayapan in K'atun 8 Ahau (1441-1460 CE). This and other historical references are increasingly corroborated by paleoclimatic, paleoenvironmental, and archeological datasets, dated using AMS [14]C and U-Th series methods. Like Mayapan's ultimate demise, drought and famine are also described in the Book of Chilam Balam of Mani (Fig. 2; 1441–1460 CE)[62].

Fisher's exact tests (one-tailed) were used to compare counts of deaths from internal conflict during Mayapan's occupation (Fig. 2). Results show significant differences between numbers of deaths in the following intervals: Early to Middle (1100–1400 cal. CE) $p = 0.053$; Early to Late (1100–1500 cal. CE) $p < 0.001$; Middle to Late (1250–1500 cal. CE) $p = 0.002$ (Fig. 2D). Population decline and violence coincided with a marked period of drying indicated by greater oxygen isotope values in the YOK-I (Belize) and nearby Chaac speleothem-based paleoclimate records, a shift to greater $\delta^{18}O$ values in snails from a sediment core taken in nearby Aguada X'caamal, and the abrupt rise in the abundance of the euryhaline benthic foraminifer *A. beccarii* in the X'caamal profile (Fig. 2; Supplementary Note 2). In the context of drought, leaders of political units who had belonged to the Mayapan confederacy returned to their home territories after the dissolution of the regional state. They headed numerous smaller polities across the peninsula, some of which were quite powerful and prosperous at the time of Spanish contact[63].

## Discussion

The period of depopulation at Mayapan also overlapped with the drought of *1 Rabbit* (1454 cal. CE) that caused severe famine in Aztec Central Mexico (Fig. 2)[64]. That famine followed closely on the heels of a devastating series of climate disasters that affected Central Mexico in 1446, especially, starting in 1450 cal. CE. Impacts of those Central Mexican droughts on the Maya region remain unclear, but surely, lucrative long-distance trade with Central Mexico would have been temporarily disrupted[65]. Trade was an especially important livelihood for a populous network of towns across the peninsula, extending from the Gulf Coast to the Caribbean. Significant fluctuations in local rainfall, especially in the form of extreme droughts, invariably affected agricultural productivity, despite the significant investment of Mayapan and its contemporary cities and towns in extensive agrarian farming, intensive cultivation of orchards and gardens, and complementary activities of hunting, fishing, and husbandry of domestic turkeys and deer[37]. Spatial differences in rainfall, soil depth, surface water, and elevation gave rise to diverse local production opportunities and histories in the Maya Lowlands[66], and importantly, differences in relative vulnerability to short-term or local climate variations. Variability in the development of water management and intensive agricultural systems also occurred in the lowlands, although such investments are generally more limited in the northwest and central Yucatán compared to other zones[67–71]. Short-lived, temporary crises were systematically addressed by exchanges

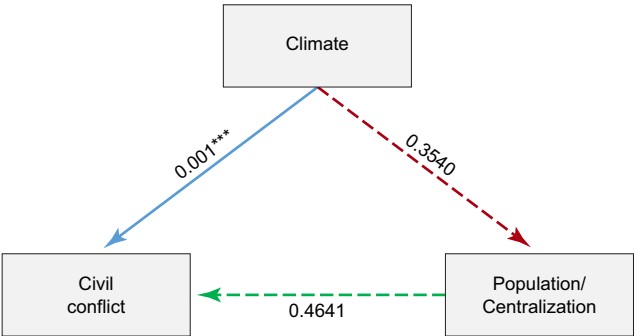

**Fig. 3 Summary of model results.** The diagram compares the direction and significance of general linear modeled relationships between climate, population, and civil conflict. *P*-values are given for each relationship, with asterisks denoting significance. Dotted lines show non-significant relationships. Colors correspond with model fit plots in Supplementary Fig. 15.

with adjacent peninsular regions of food, an important commodity sold in Pre-Columbian and Contact Period Maya marketplaces[72,73].

We used Generalized Linear Modeling (GLM) to examine the relationships among climate change, fluctuations in population size/nucleation (SPD), and osteological data indicative of internal conflict during Mayapan's entire occupational history (~1100–1450 cal. CE; Fig. 3, Supplementary Note 3, Supplementary Figs. 15-16). Osteological data from directly radiocarbon-dated human skeletal material (associated with civil conflict) were compared with local/regional climate data. Model results show a significant increase in internal conflict associated with drought conditions (Tzabnah, $p = 0.0001$; YOK-I; $p = 0.0033$; Fig. 3, Supplementary Table 2). Comparisons of climate and population size/nucleation during this same interval reveal a non-significant relationship, suggesting population dynamics were largely structured by phenomena other than local precipitation. A multivariate GLM, comparing climate records to conflict with SPD as an interaction term, rejects the hypothesis that climate only has a direct effect on internal conflict when population numbers are high.

Despite the lack of association between climate and population throughout the occupational sequence, we do find a significant relationship between drought and population decline when we examine the period between 1350 and 1430 cal. CE, i.e., the time when the most substantial population decline occurred ($p = 0.0065$; Supplementary Fig. 16). Results indicate that both climate drying and population decline or dispersal correspond to the dramatic increase in civil conflict near the end of the Mayapan occupational sequence. Finally, we also explored the relationship between warm summer/fall sea-surface temperatures in the Cariaco Basin (Venezuela)[74] and conflict, argued to be related to increased warfare in the Maya region at the end of the Classic Period (~700–900 CE)[75], but found no significant relationship ($p = 0.65$).

Our findings support Mayapan's storied institutional collapse between 1441 and 1461 cal. CE, a consequence of civil conflict driven by political rivalry and ambition, which was embedded in the social memory of Yucatecan peoples whose testimonies entered the written record of the early Colonial Period[41]. Our data indicate that institutional collapse occurred in the environmental context of drought and conflict within the city. Vulnerabilities of this coupled natural-social system existed because of the strong reliance on rain-fed maize agriculture, lack of centralized long-term grain storage, minimal opportunities for irrigation, and a sociopolitical

system led by elite families with competing political interests, from different parts of the Yucatán Peninsula. We argue that long-term, climate-caused hardships provoked restive tensions that were fanned by political actors whose actions ultimately culminated in political violence more than once at Mayapan. Direct radiocarbon dates and mitochondrial DNA sequences from the remains of individuals in the city's final mass grave suggest they were family members of the heads of state (the Cocoms), ironically and meaningfully laid to rest at the base of the Temple of K'uk'ulkan, the iconic principal temple and ritual center of Mayapan. Our results suggest that rivalry among governing elites at Mayapan materialized into action in the context of more frequent and/or severe droughts. Comparatively, such climate challenges present a range of opportunities for human actors, from the development of innovative adaptations to the stoking of revolution[5,18,76]. These climate hardships and ensuing food shortages would have undermined the city's economic base and enabled the Xiu-led usurpation. The unifying and resilient institutions that held the Mayapan state together until ~1450 CE were ultimately eroded, the confederation dissolved, and the city largely abandoned.

As a counterpart to strong archeological, historical, and climatic evidence for civil conflict and institutional collapse at Mayapan, we explored different pathways for resilience and societal transformation to regional climate change[8]. Mobility was an important component of social change as people dispersed from Mayapan to other parts of the Yucatán Peninsula[77]. This was a common indigenous response, observed historically during drought periods when Maya people abandoned Spanish towns and dispersed to settlements in the unsubdued frontier to the south[48,78,79]. The social and political landscape of the Yucatán Peninsula was also reorganized into a network of smaller states that were politically independent, but economically interwoven. Prosperous coastal towns, first observed by Spaniards,[63] took advantage of a variety of freshwater features[69,70] and marine resources that provided opportunities for subsistence and economic diversification. Maritime commerce thrived, with large canoes carrying salt, chocolate and other commodities around the peninsula[80]. At the time of European contact, the Yucatán Peninsula was divided into 15 states[63], some with developed hierarchies and kings (*halach winik*) and others more loosely organized; bureaucratic systems ranking nobles according to an array of religious and political titles[81]. The conflict between states occurred and each state had military leaders who could quickly mobilize armies, but there is little historical evidence for civil conflict within each state. Ironically, climatic impacts caused by drought and related pestilences continued in northern Yucatán with little respite in the century following Mayapan's fall[48]. Yet economic, social, and religious traditions persevered until the onset of Spanish rule, despite the reduced scale of political units, attesting to a resilient system of human-environmental adaptations[63].

Overall, we argue that human responses to drought on the Yucatán Peninsula during the 15th century CE were complex. On the one hand, drought stimulated civil conflict and institutional failure at Mayapan. However, despite decentralization, intervals of mobility, temporary impacts to trade, and political reorganization in the region, a resilient network of small Maya states persisted and was encountered by Europeans in the early 16th century[63]. These complexities are important as we attempt to evaluate the potential success or failure of modern state institutions designed to maintain internal order and peace in the face of future climate change. Recent regional climate records and modeling results suggest drier conditions in Central America are associated with global warming[82,83]. Food insecurity, social

unrest, and drought-driven migration are already of considerable concern in parts of Mexico and Central America. Our transdisciplinary work highlights the importance of understanding the complex relationships between natural and social systems, especially when evaluating the role of climate change in exacerbating internal political tensions and factionalism in areas where drought leads to food insecurity.

## Methods

The primary bioarchaeological data for this paper derives from Instituto Nacional de Antropología e Historia (INAH)-Yucatán's projects in the excavation and restoration of Mayapan's monumental zone (directed by Carlos Peraza Lope). Additional data originate from salvage archeological work conducted by Peraza Lope for INAH along the Merida-Chetumal highway that crosses through the Mayapan site, as well as National Science Foundation-supported collaborative research in the settlement zone. Permits were issued by the INAH-Consejo de Arqueología for excavation and the export of samples for analysis in the US (C.A.401-36/2172, issue date: October 21, 2009; 401-3-10492/11554, AA-53-09 A/3989, issue date: November 5, 2009; C.A.401-36/0028, issue date: January 19, 2010; C.A.401-36/1223, issue date: July 13, 2010; 401-3-7328, AA-40-10 A/ 948, issue date: August 10, 2010; 401-3-1016, AA-01-16, issue date: February 5, 2016). All work was conducted with local Yucatecan archeologists, field crews, support staff, and the community of Telchaquillo, located 1 km north of the archeological site of Mayapan. All excavation, including the excavation of human remains, was done with community permission and participation. All human remains that fall under the auspices of the Instituto Nacional de Antropología e Historia of Mexico (Yucatan) are stored at a permanent repository located at the laboratory of the Proyecto INAH Mayapán, Telchaquillo, Yucatán, Mexico. Samples associated with the Carnegie Institution of Washington excavations are curated at the laboratory of the Centro INAH Yucatán, Mérida, Yucatán, Mexico.

**Radiocarbon dating**. We report 211 AMS $^{14}C$ dates from 205 sets of human remains excavated at Mayapan (see Supplementary Data 2). Bone collagen for AMS $^{14}C$ analysis was extracted and purified using the modified Longin method with ultrafiltration[84]. Samples with low collagen yields were processed using amino acid hydrolysis and XAD purification[85]. Physically cleaned samples were demineralized and gelatinized. Crude gelatin yields were recorded and the gelatin was ultrafiltered, retaining >30 kDa molecular weight gelatin. Carbon and nitrogen concentrations and stable isotope ratios were measured at the Yale Earth Systems Center for Stable Isotopic Studies facility with a Costech elemental analyzer (ECS 4010) and a Thermo DeltaPlus Advantage isotope ratio mass spectrometer, respectively. Sample quality was evaluated by % crude gelatin yield, C%, N%, and C:N ratio. Ultrafiltered gelatin (~2.1 mg) was combusted for 3 h at 900 °C in vacuum-sealed quartz tubes with CuO and Ag wire. Graphitization and AMS $^{14}C$ measurements were done at the Keck Carbon Cycle Accelerator Mass Spectrometer facility and the Penn State University Accelerator Mass Spectrometer Laboratory. The $^{14}C$ ages were corrected for mass-dependent fractionation using measured $\delta^{13}C$ values and compared with backgrounds and known-age secondary standards. All dates were calibrated in OxCal version 4.4[86] using the IntCal20 curve[87].

**Summed probability distributions**. To estimate population centralization at Mayapan, we used the rcarbon package[88] in the R environment to generate summed probability distributions (SPDs) on calibrated radiocarbon dates from our sample of 205 dated human skeletal remains. We then compared our SPDs to a Monte-Carlo simulation of a null exponential growth model (see Supplementary Note 4) to identify general population centralization trends as well as periods of time when our SPD significantly deviated from the expected null values[89]. Higher- or lower-than-expected density of observed SPDs for a particular year indicates a local divergence of the observed SPD from the fitted exponential growth null model, and the significance of these deviations can be used to assess the goodness-of-fit using a global test.

**Generalized linear model-centralization**. We fit generalized linear models (GLMs) with a binomial distribution and log link appropriate to proportional data and quasi-likelihood estimation to account for overdispersion. To generate the GLM comparing the Mayapan SPD (response variable) to the YOK-I $\delta^{18}O$ and Tzabnah $\delta^{18}O$ data (predictor variables), we derived summed probability values and paired them with $\delta^{18}O$ values for every year both data were available. To pair SPD values with internal conflict data, we averaged values for the $2\sigma$ $^{14}C$ probability distribution for each individual. For example, if an individual had a $2\sigma$ $^{14}C$ range of AD 1100–1175, all SPD values from that temporal range were averaged. The resulting GLM was generated in the R programming environment.

**Generalized linear model-conflict**. A GLM with a binomial distribution and log link appropriate to proportional data and quasi-likelihood estimation to account for overdispersion was estimated to compare precipitation with individual deaths from civil conflict. To do this we averaged the YOK-I and Tzabnah $\delta^{18}O$ values for

the 2σ $^{14}$C probability distribution for each individual. The resulting dataset contains a sub-sample of 183 individuals, each with a death from civil conflict and YOK-I and Tzabnah δ$^{18}$O designation, which was uploaded into the R programming environment to generate the GLMs.

**Reporting summary**. Further information on research design is available in the Nature Research Reporting Summary linked to this article.

## Data availability
All source data generated in this study are provided in the Supplementary Information/ Source Data file. Osteological/Bioarchaeological data was generated at the University of New South Wales, Australia (Stanley Serafin; s.serafin@unsw.edu.au). All human remains that fall under the auspices of the Instituto Nacional de Antropología e Historia of Mexico (Yucatán) are stored at a permanent repository located at the laboratory of the Proyecto INAH Mayapan, Telchaquillo, Yucatán, Mexico. Samples associated with the Carnegie excavations are stored at the laboratory of the Centro INAH Yucatán, Mérida, Yucatán, Mexico. All sample requests should be made to INAH (Carlos Peraza; cperaza_yuc@hotmail.com). AMS radiocarbon data was produced at the Pennsylvania State University Accelerator Mass Spectrometry Facility (current contact is Brendan Culleton; bjc23@psu.edu). The speleothem and associated data were generated at Cambridge University (David Hodell; dah73@cam.ac.uk). Source data are provided with this paper.

## Code availability
All R code generated for the SPD and statistical analyses and OxCal code used for chronological modeling are available as Supplementary Data files.

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

## Acknowledgements

Much of the data for this paper derives from Instituto Nacional de Antropología e Historia (INAH)-Yucatán's investments in the excavation and restoration of Mayapan's monumental zone (directed by Carlos Peraza Lope). Additional data originate from salvage archeological work conducted by Peraza Lope along the Merida-Chetumal highway that crosses through the Mayapan site, as well as National Science Foundation-supported collaborative research in the settlement zone (NSF BCS#s 1406233, 0742128, 0109426, 1144511, and M.M.) and the National Geographic Society (NGS 8598-08). We thank the INAH-Consejo de Arqueología for granting permits for excavation and the export of samples for analysis in the US. We gratefully acknowledge the contributions of local Yucatecan archeologists, field crews, support staff, and the community of Telchaquillo, located 1 km north of the archeological site of Mayapan, without whose support this research would not have been possible. In particular, we thank Pedro Delgado Kú, Bárbara Escamilla Ojeda, Wilberth Cruz Alvarado, Luis Flores Cobá, Fernando Flores, Fernando Mena and Pancho Uc. Osteological research was financially supported by grants from Central Queensland University (S.S.; New Staff Research Grant), Middle American Research Institute of Tulane University (S.S.), and Sigma Xi (S.S.). Interdisciplinary research supported by the NSF (BCS-0940744, D.J.K) and general lab support for radiocarbon work supported by the NSF Archaeometry program (BCS-1460369, D.J.K & B.J.K.), The Pennsylvania State University (PSU; D.J.K.), and the University of California, Santa Barbara D.J.K). We are grateful to Laurie Eccles, Claire Ebert, Martin Welker, Brad Erkkila, and the interns and staff at the Human Paleoecology and Isotope Geochemistry Lab at PSU. We thank the Instituto Nacional de Antropología e Historia for granting us permission to access Cenote Ch'en Mul at Mayapan and collect a stalagmite (M1) for paleoclimate analysis. The speleothem was collected with support from National Geographic Society Grant 5917-97 to D.A.H. and M.B. and analyzed with support from the Leverhulme Trust - Research Project Grant 2019-228 to D.A.H.

## Author contributions

D.J.K., M.M., S.S., S.F.M.B., and D.A.H. designed the study. M.M., C.P.L., B.W.R., E.U.G., and E.H.P. collected archeological samples. S.S. performed the osteological analysis. D.A.H., M.B., S.M., and J.H.C. collected the Mayapan speleothem and participated in other sample collections. D.J.K., B.J.C., R.J.G., J.A.H., and T.K.H. directly radiocarbon-dated the skeletal material and established the chronology. W.C.M. generated SPD for radiocarbon dates and conducted the generalized linear modeling. D.A.H., S.F.M.B., K.M.P., T.C.S., N.M., M.Z., Y.A., V.J.P., V.V.A., S.A.C., D.H.J., A.J.M., G.H., M.B., and J.U.L.B. generated or analyzed the climate data. D.J.K. and D.A.H. supervised the study. D.J.K, M.M., S.S., S.F.M.B, W.C.M., and D.A.H. wrote the paper with contributions from all authors.

## Competing interests

The authors declare no competing interests.
