## [Peer Review File · Nature Communications]

Reviewers' Comments:

Reviewer #1:

Remarks to the Author:

This manuscript argues that prolonged drought directly caused civil conflict within the Mayan city of Mayapán, which in turn destabilized the community and led to its abandonment and collapse. The literature on climate change (megadrought) and the abandonment of Mayan cities is now extensive – especially in the late Classic period of Maya civilization – partly because this abandonment perhaps the most alarming (and therefore best-known) example of a sophisticated society (arguably) laid low by the kind of climatic trend we fear in our future.

This literature includes some impressive scholarship but – given the nature of surviving evidence – has also relied on speculation derived from crude correlation. This article, the product of evidently fruitful collaboration between a large group of interdisciplinary scholars – introduces several important innovations. First, it examines climatic trends and social responses in a considerably later period, and in a different place, than studies of the Mayan experience of climate change usually consider. Second, it presents richly detailed osteological evidence that allows it to determine how and to some extent why citizens of Mayapán died during droughts. Third, it makes use of Generalized Linear Modelling (GLM) to establish “significant relationships” and thereby causation between climatic trends and violence.

These are, to my mind, very important, wonderfully interdisciplinary innovations in an important area of scholarship with great relevance to the present and future (as the authors persuasively describe in 389-395). The authors also do a great job unpacking their methods in a detailed supplementary information. I therefore enthusiastically support publication of this article in Nature Communications.

However, I do have two broad concerns, and then some specific points regarding framing.

First, it is largely implied in this article that agricultural drought will lead to catastrophic food shortages in Mayapán. However, the local agroecosystem is never clearly described; Mayapán trade networks are only alluded to; and other wellsprings of local resilience – food stockpiles, for example – are never mentioned. This is important, because drought in fact does not necessarily reduce the amount of food available to a population; much depends on a population's subsistence strategies. These topics, therefore, require deeper engagement – especially in light of my next concern.

Second, the argument in this article is that correlations can reveal causation using the GLM method. I should stress that I am no expert in the use of GLM, and the supplementary information provided by the authors was useful to me. However, the authors provide only speculation about the causal pathways by which drought “provoked restive tensions to the surface and provoked political violence.” This is not convincing to me. Providing a more detailed description of the local/regional agroecosystem is one way to begin moving beyond such speculation. Another way might be to contextualize the cause study of Mayapán within the large (historical) scholarship on climate change and conflict, which identifies many examples of internal conflict worsened or partly provoked by climate anomalies and trends. If there are common causal mechanisms at work in these case studies, perhaps the evidence accumulated by the authors will be enough to suggest they are at play in Mayapán, too. One more option is to scrap the speculation altogether, and to simply focus on what the article has revealed: a novel collection of evidence and a novel method to begin linking climate change to violence. This unusual combination of methods and sources should be emphasized more by the authors; it has relevance far beyond Mayan case studies.

Of course, addressing these points will require lengthening the article. I found the framing of the article on its first page to be largely unconvincing, as the following notes reveal. The authors might consider removing most of this page and replacing it with several paragraphs that address the concerns I describe above.

On to the specific objections regarding the article's framing:

47-50: I would certainly not characterize the "climate conditions of the last century" as "stable"; in fact the global change in average temperature over the century is probably unprecedented since the Last Glacial Maximum. This sentence is also a bit of a simplification; the problem is that the magnitude of climate change for most of the century did not create enough stress on the societies that are usually the focus of such work compared to the socioeconomic, technological, cultural, demographic, and political changes that affected those societies (especially those associated with colonialism). Another problem is that different approaches to such research - in particular what Sam White and Qing Pei (Past Global Changes Magazine, 28(2), 44-45) identified as "cause of effect" vs "effect of cause" methods of establishing causation - are at odds with one another and so the community of scholars working on the topic cannot establish a common standard for linking climatic cause with social effect. So I would either expand this sentence to accommodate this complexity or just drop any reference to the reason for controversy about the influence of climate change on "civil conflict and societal stability."

51, 65: "relationship among" should be "relationship between."

60: I don't think the authors have found a "direct causal link" between drought and civil conflict, though I certainly believe they have found a likely link. The reason for my skepticism is that drought caused civil conflict within the preexisting socioeconomic, political, and cultural structure of Mayapán. Drought and subsistence failures do not *have* to cause conflict, but they did in this particular community - because of how that community was organized. This means that the link between drought and conflict was mediated by the structure of the community, which then suggests that it was not a direct causal link. That, in any case, is one perspective, and I suspect it will be a common one among historians at least who pursue this kind of research; the authors might want to engage with it.

65-67: The literature cited here is incomplete; some seminal studies are missing (and note that Davis, "Late Victorian Holocausts" does not consider the deep past, but rather the late nineteenth century). More on this in the next point....

67-71: This sentence is problematic. I disagree with the implication here that emphasis is more often placed - in studies that link climate change to violent conflict and collapse - on "external" warfare. Just the opposite, in my view: the argument so often made in these studies actually mirrors that of this paper. Very dry or cold conditions provoke harvest failures, which incite violence and thereby destabilize and ultimately destroy political structures. The D. Zhang et al. study cited in endnote 10 - which many in the community would view as crudely deterministic - is unusual in this regard; even some of Zhang's subsequent work examined "internal" warfare. What is missing here, above all, are citations to seminal studies or scholarly overviews - what we historians call historiographies - that reveal the magnitude of work on internal conflict and more broadly on collapse. It's obvious that key studies by for example Harvey Weiss and Geoffrey Parker should be cited. There are several scholarly overviews of the Little Ice Age (in WIRES Climate Change) that may be helpful, the same goes for "The Palgrave Handbook of Climate History." This is only scratching the surface of the surface, but it's better than nothing!

71-72: It is very strange here to see major studies by Jan Selby et al. and Marwa Daoudy not mentioned here; in the wake of these studies, it's a bit of a stretch to now characterize the link between climate change and the Syrian civil war as "controversial" (in my view, any link should now be seen as subtle if it existed at all).

86: I think the authors can find better (more recent) citations to make this point.

88-90: This is a very vague sentence (I'm also not sure it's accurate to say that IPCC assessment reports focus on "observations"); it should be reworked and probably split in two.

91-90: I don't know what is meant by "long term" in this sentence, but again the magnitude of climate change over the last 100 years is immense - especially over the last 30 years of course - and has no parallel for thousands of years.

94-95: Agricultural droughts should be defined here.

95-97: It's social *effects, and there's awkward phrasing here ("linking [climate change] effects . . . to climate change). I don't know if we can consider archaeological and historical records

“underexploited,” given the thousands of publications that now use them to explore how climate change influenced the human past; perhaps less extreme wording would be helpful here.

99: I don't think much if anything would be lost from this article if it started with this sentence and scrapped everything that came before!

112-116: Here it's mentioned as though universally accepted that “climate change” was a “contributing factor to increasing inter-group conflict” and implied that it therefore led to “sequential disintegration” of classical Mayan cities. This argument needs to be spelled out in greater detail – what exactly is the evidence? What exactly was the climate change (or variability)? Why do the authors accept the causal connection?

To conclude, I thank the authors for a stimulating article, and hope it gets published with revisions that will make it even better.

Reviewer #2:

Remarks to the Author:

The submitted ms “Drought-Induced Civil Conflict Among the Ancient Maya” is an excellent contribution to the literature on drought and its connections with ancient Maya culture. The authors represent a hall of fame of Maya archaeology and Maya and global geoscience. The technical aspects of the paper are strong, including the dating, the paleoclimatology, and osteology. These show clear evidence for severe aridity events coinciding with evidence for violence at the important Maya city of Mayapan. This city is particularly important because there are documents that provide written evidence of drought and violence from a time relatively recent to the event.

But, there should be some discussion of the reliability of written evidence since it was originally from oral transmission.

There should be some discussion of their statement “the challenges associated with constraining the uncertainties of long-term climatic, archaeological, and historical records demand a rigorous interdisciplinary framework for examining climate-society interactions”. They should explore how their ms fits the Degroot et al. criteria they cite.

They could also make the case much better (or not) for civil conflict with aDNA and SR isotope data to better identify the victims in these contexts: are they related, local, distant?

There should be some more discussion of the complexities of linking drought to collapse: a nuanced discussion of known, recent severe droughts and clear human impacts. The ms cites some fine papers but needs to consider the literature that questions these links. For example, consider the following articles by Selby and others who critique the connections:

<https://www.sciencedirect.com/science/article/pii/S0962629816301822>

https://www.sciencedirect.com/science/article/pii/S0016718518301829?casa_token=piI0Vf2U3WoAAA:cKRrHTYUMFWtqgV0wE611sTL08uudRX5Sju_IcdpQv-tIQo_rbo19ESpuSdv1vgmb747YjrlP2U

<https://link.springer.com/article/10.1007/s40641-018-0115-0>

<https://www.taylorfrancis.com/chapters/edit/10.4324/9780429446252-5/climate-change-drive-violence-conflict-human-migration-david-zhang-qing-pei-christiane-fr%C3%B6hlich-tobias-ide>

The ms characterization on lines 105-106 that “More centralized states appeared in the Maya Lowlands during the intervals 100 cal. BCE–200 cal. CE” is now superseded by the Inomata et al. papers 2020 (Nature) and 2021 Nature HB) that show earlier centralization through shared site designs and the Maya base 20 number system across a wide area around the start of the Maya culture, ~3000 bp and how many sites were similar through a wide swath of the Maya and Olmec Gulf Lowlands. This is also a very city specific discussion, but what does the evidence suggest about population trends across the surrounding area? Does the population habitat-track into wetlands or other rural

hinterlands after centers declined as some have shown for the Postclassic in recent articles (Beach et al. 2019, PNAS; Krause et al. 2021, Anthropocene)?
All said, this is a great piece of scholarship and I hope to see it published soon.

Reviewer #3:
Remarks to the Author:
(English review follows)

¿Cuáles son los resultados notables?

Siendo una especialista en bioarqueología y que he trabajado un poco el área maya, es la primera vez que leo una explicación con fundamentos científicos, desde distintas perspectivas teóricas, aportando datos duros fuera de los comúnmente aportados por la arqueología y la bioarqueología. Los resultados más notables están en el corazón del planteamiento: comprender la exacerbación de las tensiones políticas internas subyacentes en áreas donde la inseguridad alimentaria sigue a la sequía. Los autores aportan además:

El establecimiento de una secuencia de eventos, a partir de datos bioarqueológicos y fechamientos con c14 permiten hipotetizar las posible disolución del poder política y abandono de Mayapan:

- 1) Periodo de guerra de terror (1302-1323)
- 2) Periodo de persecución de los nobles Xiu (1361-1381)
- 3) Descentralización y disminución demográfica (1382-1401)
- 4) Masacre de la familia noble Cocom (1440-1461)
- 5) Disolución del poder político y abandono del sitio (después de 1450)

Estos datos fueron contrastados y apuntalados con información paleoclimática regional que permite postular la presencia de un evento de duración intermitente –periodos de sequía frecuentes-, así como con registros de estos eventos durante la época colonial y contemporánea que les permiten a los autores proponer que los periodos de sequía frecuentes propiciaron una serie de problemas sociales, falta de comida y migración.

¿Será el trabajo de importancia para el campo y el campo relacionado? ¿Cómo se compara con la literatura establecida?

Una de las grandes discusiones en el mundo académico experto en mayas tiene que ver con el colapso de esta sociedad y las hipótesis son varias, desde grandes cambios climatológicos, terremotos, guerras, sequías/inundaciones que provocaron el abandono de los grandes sitios hacia el Clásico Tardío. La extensa revisión de la bibliografía que trata el tema está resumida en este trabajo y las líneas de investigación que desarrolla están acordes con las tendencias actuales teóricas y metodológicas de este tipo de estudios. Sin embargo, este artículo provocará múltiples reacciones entre los especialistas, porque como ya lo dije antes, es la primera vez que se publica un trabajo con la intervención de múltiples especialistas: antropólogos, bioarqueólogos, geólogos, especialistas en paleoclima; esta conjunción de especialidades hace posible llegar a los resultados que se presentan, ya que desde mi punto de vista, es la única forma de llegar a una propuesta integral de un hecho que es evidente en la arquitectura, en los múltiples entierros humanos con huellas de violencia y/o desnutrición por escasez de alimentos y el abandono de un sitio.

¿El trabajo respalda las conclusiones y afirmaciones, o se necesitan pruebas adicionales?

Los archivos con información suplementaria son una verdadera joya, ellos son el soporte teórico-metodológico de esta investigación donde coinciden especialistas de distintos campos científicos. El desarrollo de trabajo y la información que vierte cada especialista son consistentes con las conclusiones y las afirmaciones sobre un posible evento de sequía, eventos intermitentes, que se acumularon en perjuicio de la población, al provocar además de carestía de alimentos e inseguridad,

una migración forzada por falta de agua.

¿Existen fallas en el análisis de datos, interpretaciones y conclusiones? ¿Esto prohíbe las publicaciones o requiere revisión?

No hay fallas, el artículo es publicable tal y como está.

¿Me suena la metodología? ¿El trabajo cumple con los estándares esperados en su campo?
Mi área de expertise es la bioarqueología, por lo que el uso de las técnicas osteológicas utilizadas para inferir periodos de violencia al interior de la población y al exterior de ella considero que es la adecuada, así como la interpretación basada en los datos de isótopos estables y carbono14.

¿Se proporcionan suficientes detalles en los métodos para que se reproduzca el trabajo?
Sí, sería conveniente publicar toda la información complementaria.

What are the notable results?

Being a specialist in bioarchaeology, I worked in the Mayan area some years ago, I've read a lot about Maya collapse, but it's the first time that I have read a scientific explanation, from different theoretical perspectives, that provides hard data outside of those commonly provided by archaeology and bioarchaeology. The most notable results are at the heart of the approach: understanding the exacerbation of underlying domestic political tensions in areas where food insecurity follows drought. The authors also contribute:

The establishment of a sequence of events, based on bioarchaeological data and dating with c14 allows hypothesizing the possible dissolution of political power and abandonment of Mayapan:

- 1) Period of war of terror (1302-1323)
- 2) Period of persecution of the Xiu nobles (1361-1381)
- 3) Decentralization and demographic decline (1382-1401)
- 4) Massacre of the noble family Cocom (1440-1461)
- 5) Dissolution of political power and abandonment of the siege (after 1450)

These data were contrasted and supported with regional palaeoclimatic information that allows consideration of the presence of an event of intermittent duration - frequent drought periods - in comparison with records of historical events during the colonial and contemporary times. This approach allows the authors to propose that frequent dry spells led to a series of social problems, including lack of food and migration.

Will the work be relevant to the field and related field? How does it compare to the established literature?

One of the important academic discussions about Mayans has to do with the collapse of society. Current hypotheses vary, from great climatic changes, to earthquakes, wars, and droughts / floods that caused the abandonment of the great sites towards Late Classic period. Extensive bibliographic review about this debate is summarized in this work. The lines of research discussed are in accordance with the current theoretical and methodological trends of this type of study. However, this article will provoke a response among specialists, because as I said before, it is the first time that a work has been published with the involvement of multiple specialists: anthropologists, bioarchaeologists, geologists, and paleoclimate specialists. This conjunction of specialties facilitates the results that are presented. From my point of view, this is the only way to arrive at a comprehensive proposal of a fact that is evident in architecture, in the multiple human burials with traces of violence, and / or malnutrition due to food shortages and the abandonment of a site.

Does the work support the conclusions and claims, or is additional evidence needed?

The files with supplementary information are a true gem. They are the theoretical-methodological support of this research, where specialists from different scientific fields cooperate.

The development of the manuscript and the information provided by each specialist are consistent with the conclusions and statements about a possible drought event, intermittent events, which accumulated to the detriment of the population, by causing, in addition to food shortages and insecurity, a migration forced due to lack of water.

Are there flaws in the data analysis, interpretations, and conclusions? Does this prohibit postings or require review?

There are no flaws, the article is publishable as it is.

Does the methodology sound familiar to me? Does the work meet the standards expected in your field?

My area of expertise is bioarchaeology. I consider the use of osteological techniques to infer periods of violence within the population and outside of it to be adequate, as well as the interpretation based on stable isotope data and carbon14.

Are sufficient details provided in the methods for the work to be reproduced?

Yes, it would be convenient to publish all the supplementary information.

Point-by-Point Responses

Many thanks to the reviewers for the positive feedback. We appreciate their careful review and suggestions. Below we address the comments made by the three reviewers. These comments have helped clarify key points and we have added new text in red to the main manuscript. There were no substantive changes to the supplement based on the reviews.

REVIEWER 1

Reviewer #1, Comment 1

This manuscript argues that prolonged drought directly caused civil conflict within the Mayan city of Mayapán, which in turn destabilized the community and led to its abandonment and collapse. The literature on climate change (megadrought) and the abandonment of Mayan cities is now extensive – especially in the late Classic period of Maya civilization – partly because this abandonment perhaps the most alarming (and therefore best-known) example of a sophisticated society (arguably) laid low by the kind of climatic trend we fear in our future.

This literature includes some impressive scholarship but – given the nature of surviving evidence – has also relied on speculation derived from crude correlation. This article, the product of evidently fruitful collaboration between a large group of interdisciplinary scholars – introduces several important innovations. First, it examines climatic trends and social responses in a considerably later period, and in a different place, than studies of the Mayan experience of climate change usually consider. Second, it presents richly detailed osteological evidence that allows it to determine how and to some extent why citizens of Mayapán died during droughts. Third, it makes use of Generalized Linear Modelling (GLM) to establish “significant relationships” and thereby causation between climatic trends and violence.

These are, to my mind, very important, wonderfully interdisciplinary innovations in an important area of scholarship with great relevance to the present and future (as the authors persuasively describe in 389-395). The authors also do a great job unpacking their methods in a detailed supplementary information. I therefore enthusiastically support publication of this article in Nature Communications.

Reviewer 1, Response 1. We appreciate these comments and thank the reviewer for the support. We address the more specific comments below.

Reviewer #1, Comment 2

First, it is largely implied in this article that agricultural drought will lead to catastrophic food shortages in Mayapán. However, the local agroecosystem is never clearly described; Mayapán trade networks are only alluded to; and other wellsprings of local

resilience – food stockpiles, for example – are never mentioned. This is important, because drought in fact does not necessarily reduce the amount of food available to a population; much depends on a population's subsistence strategies. These topics, therefore, require deeper engagement – especially in light of my next concern.

Reviewer #1, Response 2. We have provided more details about the agroecosystem, subsistence strategies, diet, trade, and storage throughout the manuscript. We emphasize the focal importance of maize as a staple grain and also the limitations of grain storage in the Yucatan. We also provide historical examples of the linkages between drought, reduced maize productivity from the immediate area.

Examples of changes include:

“The magnitude and duration of the climate impacts in the northern Yucatán (Mayapán) may have transcended established mechanisms to overcome modest inter-annual rainfall fluctuations (Fedick and Santiago 2022), given the limitations of long-term storage of maize, the primary staple grain (Kennett et al. 2012, 2020).”

“.....which was once home to 15-20,000 inhabitants who were sustained by forest gardens, hunting, and rainfed maize agriculture, supplemented by trade Masson and Peraza Lope 2014). Dietary stable isotope studies indicate heavy reliance on maize, a crop that was highly sensitive to periodic droughts, given the limitations for long-term grain storage (Kennett 2016; George 2020).”

“Farming in the region depends heavily on the timing of the onset, duration, and total effective rainfall of the wet season (Madero et al. 2020). Although short-term droughts tend to be spatially and temporally patchy in northern Yucatán (De la Barreda et al. 2020), more protracted droughts during the historic period are evident in documentary and paleoclimate archives (Hoggarth et al. 2017; Mendoza et al. 2007). Those extended dry intervals of the Colonial period caused well-documented crop failures, severe famines, and high mortality that destabilized the economy and led to dispersal of populations from major urban areas in the northern Yucatán, within the vicinity of Mayapán (Hoggarth et al. 2017).”

“Trade was an especially important livelihood for a populous network of towns across the peninsula, extending from the Gulf Coast to the Caribbean.”

“Spatial differences in rainfall, soil depth, and surface water and elevation gave rise to diverse local production opportunities and histories in the Maya Lowlands (Garrison et al. 2019) and importantly, differences in relative vulnerability to short-term or local climate variations. Variability in the development of water management and intensive agricultural systems also occurred in the lowlands, although such investments are generally more limited in the northwest and central Yucatán compared to other zones (Tankersley et al. 2020; Scarborough and Sierra 2015; Beach et al. 2019; Kennett and Beach 2013). Short-lived, temporary crises were systematically addressed by exchanges with adjacent peninsular regions for food, an important commodity sold in Pre-Columbian and Contact Period Maya marketplaces....”

“Vulnerabilities of this coupled natural-social system existed because of the strong reliance on rain-fed maize agriculture, lack of centralized long-term grain storage, minimal investments in irrigation, and a sociopolitical system led by elite families with competing political interests, from different parts of the Yucatán Peninsula.”

Reviewer #1, Comment 3

Second, the argument in this article is that correlations can reveal causation using the GLM method. I should stress that I am no expert in the use of GLM, and the supplementary information provided by the authors was useful to me. However, the authors provide only speculation about the causal pathways by which drought “provoked restive tensions to the surface and provoked political violence.” This is not convincing to me. Providing a more detailed description of the local/regional agroecosystem is one way to begin moving beyond such speculation. Another way might be to contextualize the cause study of Mayapán within the large (historical) scholarship on climate change and conflict, which identifies many examples of internal conflict worsened or partly provoked by climate anomalies and trends. If there are common causal mechanisms at work in these case studies, perhaps the evidence accumulated by the authors will be enough to suggest they are at play in Mayapán, too. One more option is to scrap the speculation altogether, and to simply focus on what the article has revealed: a novel collection of evidence and a novel method to begin linking climate change to violence. This unusual combination of methods and sources should be emphasized more by the authors; it has relevance far beyond Mayan case studies.

Of course, addressing these points will require lengthening the article. I found the framing of the article on its first page to be largely unconvincing, as the following notes reveal. The authors might consider removing most of this page and replacing it with several paragraphs that address the concerns I describe above.

Reviewer # 1, Response 3.

We have greatly modified the introduction in response to these excellent comments and we have emphasized our transdisciplinary approach in the abstract (and introduction) and our emphasis on human agency in the context of complex natural and human systems.

For example, we have added the following:

Page 2: “.....and an important nexus for debate revealing the importance of human agency and unexpected, non-linear relationships between climate and human behavior”

Page 2: “Longer-term climatic, archaeological, and historical records can contribute to these contemporary debates, but demand a rigorous transdisciplinary framework that bridges natural and social systems.”

As we outlined in our response to Comment #2 we have provided a more detailed description of the regional agroecosystem and we have emphasized the novel

combination of methods to build the case for a causal pathway between drought, conflict and sociopolitical reorganization in the northern Yucatan.

Reviewer #1, Comment 4

47-50: I would certainly not characterize the "climate conditions of the last century" as "stable"; in fact the global change in average temperature over the century is probably unprecedented since the Last Glacial Maximum. This sentence is also a bit of a simplification; the problem is that the magnitude of climate change for most of the century did not create enough stress on the societies that are usually the focus of such work compared to the socioeconomic, technological, cultural, demographic, and political changes that affected those societies (especially those associated with colonialism). Another problem is that different approaches to such research - in particular what Sam White and Qing Pei (Past Global Changes Magazine, 28(2), 44-45) identified as "cause of effect" vs "effect of cause" methods of establishing causation - are at odds with one another and so the community of scholars working on the topic cannot establish a common standard for linking climatic cause with social effect. So I would either expand this sentence to accommodate this complexity or just drop any reference to the reason for controversy about the influence of climate change on "civil conflict and societal stability."

Reviewer #1, Response 4. We have dropped the statement about the stability of climate conditions in the last century and we have expanded the introduction to frame the complexities of the debate and the importance of human agency and non-linear relationships between climate and human behavior.

For example:

".....and an important nexus for debate revealing the importance of human agency and unexpected, non-linear relationships between climate and human behavior...."

Reviewer #1, Comment 5

51, 65: "relationship among" should be "relationship between."

Reviewer #1, Response 5

We have made this change.

Reviewer #1, Comment 6

60: I don't think the authors have found a "direct causal link" between drought and civil conflict, though I certainly believe they have found a likely link. The reason for my skepticism is that drought caused civil conflict within the preexisting socioeconomic, political, and cultural structure of Mayapán. Drought and subsistence failures do not *have* to cause conflict, but they did in this particular community – because of how that community was organized. This means that the link between drought and conflict was mediated by the structure of the community, which then suggests that it was not a direct causal link. That, in any case, is one perspective, and I suspect it will be a common one among historians at least who pursue this kind of research; the authors might want to engage with it.

Reviewer #1, Response 6

We have made this word change and backed off on the idea that this is a direct causal link, which was a model outcome that we over-emphasized in the first draft. Much of this section is now focused on providing more contextual details regarding our work at Mayapán as an example of transdisciplinary research designed to explore civil conflict given the complexities of natural and social systems in the northern Yucatan.

Reviewer #1, Comment 7

65-67: The literature cited here is incomplete; some seminal studies are missing (and note that Davis, "Late Victorian Holocausts" does not consider the deep past, but rather the late nineteenth century). More on this in the next point....

Reviewer #1, Response 7

We have rewritten this section and added several key references here as suggested by the reviewer (Parker, Weiss)

Reviewer #1, Comment 8

67-71: This sentence is problematic. I disagree with the implication here that emphasis is more often placed – in studies that link climate change to violent conflict and collapse

– on “external” warfare. Just the opposite, in my view: the argument so often made in these studies actually mirrors that of this paper. Very dry or cold conditions provoke harvest failures, which incite violence and thereby destabilize and ultimately destroy political structures. The D. Zhang et al. study cited in endnote 10 – which many in the community would view as crudely deterministic – is unusual in this regard; even some of Zhang’s subsequent work examined “internal” warfare. What is missing here, above all, are citations to seminal studies or scholarly overviews – what we historians call historiographies – that reveal the magnitude of work on internal conflict and more broadly on collapse. It’s obvious that key studies by for example Harvey Weiss and Geoffrey Parker should be cited. There are several scholarly overviews of the Little Ice Age (in WIREs Climate Change) that may be helpful, the same goes for "The Palgrave Handbook of Climate History." This is only scratching the surface of the surface, but it’s better than nothing!

Reviewer #1, Response 8

We have dropped the discussion of internal vs. external warfare (see Introduction) and the false dichotomy that it created in the previous draft. We have also cited the seminal studies suggested by the reviewer (Parker, Weiss, Degroot).

Reviewer #1, Comment 9

71-72: It is very strange here to see major studies by Jan Selby et al. and Marwa Daoudy not mentioned here; in the wake of these studies, it’s a bit of a stretch to now characterize the link between climate change and the Syrian civil war as “controversial” (in my view, any link should now be seen as subtle if it existed at all).

Reviewer #1, Response 9

We have dropped specific reference to the debates surrounding drought and the Syrian civil war and we now cite Selby's work more generally.

Reviewer #1, Comment 10

86: I think the authors can find better (more recent) citations to make this point.

Reviewer #1, Response 10

We have rewritten this section as a set-up for the Mayapan test case, so this section no longer exists.

Reviewer #1, Comment 11

88-90: This is a very vague sentence (I'm also not sure it's accurate to say that IPCC assessment reports focus on "observations"); it should be reworked and probably split in two.

Reviewer #1, Response 11

We have changed this to read:

"Long-term climate reconstructions reveal that some climate variations in the past were of significantly greater magnitude than those experienced in the last 100 years. Furthermore, current and future anthropogenic influences are projected to amplify the severity of extreme events in the water cycle and lead to more intense and prolonged droughts impacting agricultural productivity than those occurring in the recent past."

Reviewer #1, Comment 12

91-90: I don't know what is meant by "long term" in this sentence, but again the magnitude of climate change over the last 100 years is immense – especially over the last 30 years of course – and has no parallel for thousands of years.

Reviewer #1, Response 12

See Response #11 for rewrite of this section.

Reviewer #1, Comment 13

94-95: Agricultural droughts should be defined here.

Reviewer #1, Response 13

Rewritten as ".....droughts impacting agricultural productivity..."

Reviewer #1, Comment 14

95-97: It's social *effects, and there's awkward phrasing here ("linking [climate change] effects . . . to climate change). I don't know if we can consider archaeological and historical records "underexploited," given the thousands of publications that now use them to explore how climate change influenced the human past; perhaps less extreme wording would be helpful here.

Reviewer #1, Response 14

We have rewritten the introduction and the only reference to archaeological records occurs in the first paragraph:

“Longer-term climatic, archaeological, and historical records can contribute to these contemporary debates, but demand a rigorous transdisciplinary framework that bridges natural and social systems.”

Reviewer #1, Comment 15

99: I don't think much if anything would be lost from this article if it started with this sentence and scrapped everything that came before!

Reviewer #1, Response 15

We have completely rewritten the introduction in response to this comment (see manuscript).

Reviewer #1, Comment 16

112-116: Here it's mentioned as though universally accepted that “climate change” was a “contributing factor to increasing inter-group conflict” and implied that it therefore led to “sequential disintegration” of classical Mayan cities. This argument needs to be spelled out in greater detail – what exactly is the evidence? What exactly was the climate change (or variability)? Why do the authors accept the causal connection?

Reviewer #1, Response 16

We have expanded this discussion to emphasize the cultural history of the Maya region in order to provide broader historical context for the Mayapan test case. In this summary we mention with argument that drought and conflict were possible contributing factors for the sociopolitical transformations evident at the end of the Classic period and we provide references to these arguments.

Reviewer #1, Comment 17

To conclude, I thank the authors for a stimulating article, and hope it gets published with revisions that will make it even better.

Reviewer #1, Comment 17

We thank the reviewer for the support and the excellent comments that have improved the manuscript tremendously.

REVIEWER 2

Reviewer #2, Comment 1

The submitted ms “Drought-Induced Civil Conflict Among the Ancient Maya” is an excellent contribution to the literature on drought and its connections with ancient Maya culture. The authors represent a hall of fame of Maya archaeology and Maya and global

geoscience. The technical aspects of the paper are strong, including the dating, the paleoclimatology, and osteology. These show clear evidence for severe aridity events coinciding with evidence for violence at the important Maya city of Mayapan. This city is particularly important because there are documents that provide written evidence of drought and violence from a time relatively recent to the event.

Reviewer #2, Response 1: We appreciate these comments and thank the reviewer for the support. We address the more specific comments below.

Reviewer #2, Comment 2

There should be some discussion of the reliability of written evidence since it was originally from oral transmission.

Reviewer #2, Response 2:

We now highlight the fact that historical records have biases and how we are verifying/calibrating these records using archaeological, osteological and radiocarbon data:

“Historical records can be biased so we used archaeological, osteological, and radiocarbon data, to establish the timing of a series of events recorded historically using the Maya K’atun calendar”

Reviewer #2, Comment 3

There should be some discussion of their statement “the challenges associated with constraining the uncertainties of long-term climatic, archaeological, and historical records demand a rigorous interdisciplinary framework for examining climate-society interactions”. They should explore how their ms fits the Degroot et al. criteria they cite.

Reviewer #2, Response 2:

We have rewritten the introduction with greater focus on the transdisciplinary research promoted by Degroot et al. and we have added a paragraph at the end of the manuscript to engage with the transformational themes highlighted by Degroot:

“As a counterpart to strong archaeological, historical, and climatic evidence for civil conflict and institutional collapse at Mayapán, we explored different pathways for resilience and societal transformation to regional climate change (Degroot et al. 2021). Mobility was an important component of social change as people dispersed from Mayapán to other parts of the Yucatán Peninsula (Rice 2009). This was a common indigenous response, observed historically during drought periods when Maya people abandoned Spanish towns and dispersed into surrounding forests (Hoggarth et al. 2017; Jones 1990; Restall 1995). The social and political landscape of the Yucatán Peninsula was also reorganized into a network of smaller states that were politically independent but economically interwoven. Prosperous coastal towns, first observed by Spaniards, (Marilyn A. Masson 2021) took advantage of a variety of freshwater features (Beach et al. 2019; Krause et al. 2021), marine resources that provided opportunities for subsistence and economic diversification. Maritime commerce thrived,

with large canoes carrying salt, chocolate and other commodities around the peninsula (Alpuche, McAnany, and Dedrick 2020). At the time of European contact, the Yucatán Peninsula was divided into 15 states (Marilyn A. Masson 2021), some with developed hierarchies and kings (*halach winik*) and others more loosely organized; bureaucratic systems organized nobles according to an array of religious and political titles (Restall 2001). Conflict between states occurred and each state had military leaders that could quickly mobilize armies, but there is little historical evidence for civil conflict within each state. Ironically, climatic impacts due to drought and related pestilences continued in northern Yucatán with little respite in the century following Mayapán's fall (Hoggarth et al. 2017). Yet economic, social, and religious traditions persevered until the onset of Spanish rule, despite the reduced scale of political units, attesting to a resilient system of human-environmental adaptations (Marilyn A. Masson 2021).”

Reviewer #2, Comment 4

They could also make the case much better (or not) for civil conflict with aDNA and SR isotope data to better identify the victims in these contexts: are they related, local, distant?

Reviewer #2, Response 4

We now make reference to the available aDNA and strontium isotope data for Mayapan. Both support our case for civil conflict. As expected based on historical accounts, strontium data indicates that the city was comprised of local and non-local individuals from other parts of the Yucatan (and sometimes farther afield). The aDNA evidence indicates that the mass burials argued to be the result of civil conflict are comprised of related individuals. Details of this work will be the subject of future publications.

Examples in the text:

Page 4: “Strontium isotopes in human teeth indicate that population aggregation and recruitment from throughout the Yucatán Peninsula, and occasionally beyond, persisted throughout Mayapán’s history (George 2020).”

Page 9: “Mitochondrial DNA indicates that these individuals shared strong maternal genetic affinities and were potentially related on the maternal line (George 2020).”

Reviewer #2, Comment 5

There should be some more discussion of the complexities of linking drought to collapse: a nuanced discussion of known, recent severe droughts and clear human impacts. The ms cites some fine papers but needs to consider the literature that questions these links. For example, consider the following articles by Selby and others who critique the connections:

<https://www.sciencedirect.com/science/article/pii/S0962629816301822>

https://www.sciencedirect.com/science/article/pii/S0016718518301829?casa_token=pil

[0Vf2U3WoAAAAA:cKRrHTYUMFWtqgV0wE611sTL08uudRX5Sju_lcdpQv-tlQo_rbo19ESpuSdv1vgmb747YjrlP2U](https://doi.org/10.1007/s40641-018-0115-0)

<https://link.springer.com/article/10.1007/s40641-018-0115-0>

<https://www.taylorfrancis.com/chapters/edit/10.4324/9780429446252-5/climate-change-drive-violence-conflict-human-migration-david-zhang-qing-pei-christiane-fr%C3%B6lich-tobias-ide>

Reviewer #2, Response 5

We have rewritten the introduction to highlight the complexities of linking drought, conflict, and transformation (collapse) and cite the literature proposed by Reviewer #2.

Reviewer #2, Comment 6

The ms characterization on lines 105-106 that “More centralized states appeared in the Maya Lowlands during the intervals 100 cal. BCE–200 cal. CE” is now superseded by the Inomata et al. papers 2020 (Nature) and 2021 Nature HB) that show earlier centralization through shared site designs and the Maya base 20 number system across a wide area around the start of the Maya culture, ~3000 bp and how many sites were similar through a wide swath of the Maya and Olmec Gulf Lowlands.

Reviewer #2, Response 6

We have cited these papers and rewritten this section.

Reviewer #2, Comment 7

This is also a very city specific discussion, but what does the evidence suggest about population trends across the surrounding area? Does the population habitat-track into wetlands or other rural hinterlands after centers declined as some have shown for the Postclassic in recent articles (Beach et al. 2019, PNAS; Krause et al. 2021, Anthropocene)?

Reviewer #2, Response 7

We have added text throughout the manuscript that provides greater environmental and sociopolitical context and cited these important references:

For example:

“Spatial differences in rainfall, soil depth, and surface water and elevation gave rise to diverse local production opportunities and histories in the Maya Lowlands (Garrison, Houston, and Alcover Firpi 2019), and importantly, differences in relative vulnerability to short-term or local climate variations. Variability in the development of water management and intensive agricultural systems also occurred in the lowlands, although such investments are generally more limited in the northwest and central Yucatán compared to other zones (Tankersley et al. 2020; Scarborough and Sierra 2015; Beach et al. 2019; Krause et al. 2021; D. J. Kennett and Beach 2013). Short-lived, temporary crises were systematically addressed by exchanges with adjacent peninsular regions for

food, an important commodity sold in Pre-Columbian and Contact Period Maya marketplaces(Freidel and Shaw 2000; Marilyn A. Masson and Freidel 2012).

Reviewer #2, Comment 8

All said, this is a great piece of scholarship and I hope to see it published soon.

Reviewer #2, Response 8

We appreciate these comments and thank the reviewer for the support.

REVIEWER 3

Reviewer #3, Comment 1

What are the notable results?

Being a specialist in bioarchaeology, I worked in the Maya area some years ago, I've read a lot about Maya collapse, but it's the first time that I have read a scientific explanation, from different theoretical perspectives, that provides hard data outside of those commonly provided by archaeology and bioarchaeology. The most notable results are at the heart of the approach: understanding the exacerbation of underlying domestic political tensions in areas where food insecurity follows drought. The authors also contribute:

The establishment of a sequence of events, based on bioarchaeological data and dating with c14 allows hypothesizing the possible dissolution of political power and abandonment of Mayapan:

- 1) Period of war of terror (1302-1323)
 - 2) Period of persecution of the Xiu nobles (1361-1381)
 - 3) Decentralization and demographic decline (1382-1401)
 - 4) Massacre of the noble family Cocom (1440-1461)
 - 5) Dissolution of political power and abandonment of the siege (after 1450)
- These data were contrasted and supported with regional palaeoclimatic information that allows consideration of the presence of an event of intermittent duration - frequent drought periods – in comparison with records of historical events during the colonial and contemporary times. This approach allows the authors to propose that frequent dry spells led to a series of social problems, including lack of food and migration.

Reviewer #3, Response 1

Reviewer #3 is a bioarchaeologist with experience in the Maya area. They point out that we address a significant problem, that of Maya collapse, using a novel approach incorporating paleoclimatology, radiocarbon dating, archaeology and bioarchaeology. They also point out its broader relevance to research attempting to understand how drought can stress and lead to irrevocable changes in political systems. They appreciate how the identification of the physical traces of historical events permitted us to reconstruct the historical processes that lead from drought to migration, food

insecurity and ever-worsening internal political tensions. We appreciate this strongly positive feedback.

Reviewer #3, Response 2

Will the work be relevant to the field and related field? How does it compare to the established literature?

One of the important academic discussions about Mayans has to do with the collapse of society. Current hypotheses vary, from great climatic changes, to earthquakes, wars, and droughts / floods that caused the abandonment of the great sites towards Late Classic period. Extensive bibliographic review about this debate is summarized in this work. The lines of research discussed are in accordance with the current theoretical and methodological trends of this type of study. However, this article will provoke a response among specialists, because as I said before, it is the first time that a work has been published with the involvement of multiple specialists: anthropologists, bioarchaeologists, geologists, and paleoclimate specialists. This conjunction of specialties facilitates the results that are presented. From my point of view, this is the only way to arrive at a comprehensive proposal of a fact that is evident in architecture, in the multiple human burials with traces of violence, and / or malnutrition due to food shortages and the abandonment of a site.

Reviewer #3, Response 2

Reviewer #3 cites the novelty of synthesizing data from diverse fields including anthropology, bioarchaeology, geology and paleoclimatology. They stress that only this sort of multidisciplinary endeavor is capable of addressing a problem as complex as Maya collapse. We appreciate this positive feedback.

Reviewer #3, Comment 3

Does the work support the conclusions and claims, or is additional evidence needed?

The files with supplementary information are a true gem. They are the theoretical-methodological support of this research, where specialists from different scientific fields cooperate.

The development of the manuscript and the information provided by each specialist are consistent with the conclusions and statements about a possible drought event, intermittent events, which accumulated to the detriment of the population, by causing, in addition to food shortages and insecurity, a migration forced due to lack of water.

Reviewer #3, Response 3

Reviewer #3 refers to our Supplementary Information as “a true gem” and underlines that the data and analyses provided by different specialists are comprehensive and fit together to support our conclusions.

Reviewer #3, Comment 4

Are there flaws in the data analysis, interpretations, and conclusions? Does this prohibit postings or require review?

There are no flaws, the article is publishable as it is.

Reviewer #3, Response 4

Reviewer #3, concludes that our study is free from errors and is publishable in its current form.

Reviewer #3, Comment 5

Does the methodology sound familiar to me? Does the work meet the standards expected in your field?

My area of expertise is bioarchaeology. I consider the use of osteological techniques to infer periods of violence within the population and outside of it to be adequate, as well as the interpretation based on stable isotope data and carbon14.

Reviewer #3, Response 5

Reviewer #3 is a bioarchaeologist and indicates that the osteological analyses and radiocarbon dating meet the standards expected in the field.

Reviewer #3, Comment 6

Are sufficient details provided in the methods for the work to be reproduced?

Yes, it would be convenient to publish all the supplementary information.

Reviewer #3, Response 6

Reviewer #3 indicates that the data and analyses provided are comprehensive.

Reviewers' Comments:

Reviewer #1:

Remarks to the Author:

This article, to my mind, is now the most significant and compelling that has been written on the connection between climatic change and the pre-colonial Maya. That is really saying something, because few if any topics have received as much attention among scholars who link climatic and human histories as the fate of the Maya. The authors have engaged thoughtfully with suggestions provided by their peer reviewers - including my own - and made revisions that have meaningfully improved an already impressive paper. To my mind this revised version is absolutely ready to be published in Nature Communications. It is a joy to read.

I have only three very minor recommendations for the first pages, which can be addressed by changing just a few words:

80: there's a tense change here (I think "examined" should be "examine").

85: "evaluated" should probably be "evaluate."

93: this statement is absolutely true, but it might give the impression that archaeological and historical records are **only** suited to examine societal responses to climate change "over a century or more."

Reviewer #2:

Remarks to the Author:

This revision is superb and better than I had even hoped for. This paper is now a major, important contribution. This is a paper that handles the drought and societal impacts in a past society as well as any I have seen with excellent climate records, chronology, and multiple lines of cutting edge and traditional evidence. This also handles the literature on modern droughts and societal impacts judiciously, making the connections and correlations but not overselling direct causes. Thus, I am convinced by these lines of evidence and careful reading of the archaeological record to suggest significant drought impacts on Postclassic Maya history in and around Mayapan and beyond.

Some suggested changes:

line 98 to start of 99, replace existing clause with "institutions developed to cope with memorable timeframes of events that were near enough to enter oral histories."

Lines 101-102: given the limitations of long-term storage of maize, the primary staple grain" add problem of distant and difficult transportation

Line 124, rather than drought-related . . . , suggest: "political and demographic decline that coincide with multiple lines of evidence for drought during the eleventh century cal. CE.

Lines 129-134 might work better in line 122 before "Northern centers . . . where the Late Classic discussion is. As is, it seems out of place through this discussion.

Lines 157-Change "Strontium isotopes in human teeth indicate that population aggregation and recruitment from throughout the Yucatán Peninsula, and occasionally beyond, persisted throughout Mayapán's history" to "Population aggregation and recruitment from throughout the Yucatán Peninsula, and occasionally beyond, persisted throughout Mayapán's history as indicated by strontium isotopes in human teeth."

Line 200, add "and distance decay from the Caribbean water source" after "easterly winds"

Point-by-Point Response

Reviewer #1 (Remarks to the Author):

This article, to my mind, is now the most significant and compelling that has been written on the connection between climatic change and the pre-colonial Maya. That is really saying something, because few if any topics have received as much attention among scholars who link climatic and human histories as the fate of the Maya. The authors have engaged thoughtfully with suggestions provided by their peer reviewers - including my own - and made revisions that have meaningfully improved an already impressive paper. To my mind this revised version is absolutely ready to be published in Nature Communications. It is a joy to read.

I have only three very minor recommendations for the first pages, which can be addressed by changing just a few words:

80: there's a tense change here (I think "examined" should be "examine").

We have changed this tense.

85: "evaluated" should probably be "evaluate."

We have changed this tense.

93: this statement is absolutely true, but it might give the impression that archaeological and historical records are **only** suited to examine societal responses to climate change "over a century or more."

We have removed the phrase "over a century or more."

Response, Reviewer #1: We appreciate these comments and thank the reviewer for the support. Small changes have been made on the current manuscript.

Reviewer #2 (Remarks to the Author):

This revision is superb and better than I had even hoped for. This paper is now a major, important contribution. This is a paper that handles the drought and societal impacts in a past society as well as any I have seen with excellent climate records, chronology, and multiple lines of cutting edge and traditional evidence. This also handles the literature on modern droughts and societal impacts judiciously, making the connections and correlations but not overselling direct causes. Thus, I am convinced by these lines of evidence and careful reading of the archaeological record to suggest significant drought impacts on Postclassic Maya history in and around Mayapan and beyond.

Some suggested changes:

line 98 to start of 99, replace existing clause with "institutions developed to cope with memorable timeframes of events that were near enough to enter oral histories."

We have added this in red text.

Lines 101-102: given the limitations of long-term storage of maize, the primary staple grain" add problem of distant and difficult transportation

We have added this in red text.

Line 124, rather than drought-related . . . , suggest: "political and demographic decline that coincide with multiple lines of evidence for drought during the eleventh century cal. CE.

We have added this in red text:

Lines 129-134 might work better in line 122 before "Northern centers . . . where the Late Classic discussion is. As is, it seems out of place through this discussion.

We have shifted this text and reorganized to read:

The Lowlands region was dotted with kingdoms large and small, amidst a densely populated countryside of towns and villages throughout the Classic period with the Late Classic Period (600–800 cal. CE) witnessing the emergence of the largest number of competing states in the southern lowlands. Shifting centers of influence and the most dramatic sequential disintegration of these cities during the Terminal Classic period, between 800 and 1000 cal. CE, with climate change argued to be a contributing factor to increasing conflict^{2,35–38}.

Lines 157-Change "Strontium isotopes in human teeth indicate that population aggregation and recruitment from throughout the Yucatán Peninsula, and occasionally beyond, persisted throughout Mayapán's history" to "Population aggregation and recruitment from throughout the Yucatán Peninsula, and occasionally beyond, persisted throughout Mayapán's history as indicated by strontium isotopes in human teeth."

We have made this change in red text.

Line 200, add "and distance decay from the Caribbean water source" after "easterly winds"

We have added this text in red.

Response, Reviewer #2: We appreciate these comments and thank the reviewer for the support. We have made all of these changes and new text is added in red.